# Fast and light-efficient remote focusing for volumetric voltage imaging

Urs L. Böhm [1,2] & Benjamin Judkewitz [1] ✉

Voltage imaging holds great potential for biomedical research by enabling noninvasive recording of the electrical activity of excitable cells such as neurons or cardiomyocytes. Camera-based detection can record from hundreds of cells in parallel, but imaging entire volumes is limited by the need to focus through the sample at high speeds. Remote focusing techniques can remedy this drawback, but have so far been either too slow or light-inefficient. Here, we introduce flipped image remote focusing, a remote focusing method that doubles the light efficiency compared to conventional beamsplitter-based techniques and enables high-speed volumetric voltage imaging at 500 volumes/s. We show the potential of our approach by combining it with light sheet imaging in the zebrafish spinal cord to record from >100 spontaneously active neurons in parallel.

Functional imaging with genetically encoded indicators of neural activity can reveal detailed insights into the inner workings of the nervous system. Light sheet imaging in combination with genetically encoded voltage indicators has shown great potential to record the membrane potential of tens of neurons in parallel[1,2], but due to the high acquisition speeds necessary for voltage sensors (500–1000 Hz) imaging is usually limited to a single focal plane. One major reason for this limitation is the requirement to focus through the sample at sufficient speed. Since most neural tissue exists in 3D, it is desirable to have microscopy methods that can image from a volume instead of a single focal plane.

A common technique for focusing through a sample in light sheet microscopy is to move the imaging objective axially with a piezo-electric actuator, but due to the inertia of relatively heavy objectives this is limited to a few tens of Hz at best. An alternative approach is to use remote focusing, which places the focusing element away from the primary objective and the sample. Remote focusing can be achieved in two ways: either by introducing defocus in a Fourier plane of the imaging system by using a tunable focusing element[3–7] or by refocusing a remote image plane (remote focusing)[8–10]. The latter method has the advantage that it can be used to achieve fast aberration-free imaging over a relatively wide z-range away from the nominal focal plane of the primary objective[8]. However, by using a small movable mirror in the remote image plane (Fig. 1a, left), this method has the disadvantage of losing >50% of light due to the necessity of a quarter-wave plate and a

polarizing beamsplitter (PBS) to separate incoming and refocused light. This is especially harmful in ultra-high-speed recordings such as voltage imaging, where integration times are in the sub-millisecond range and the signal-to-noise ratio (SNR) is severely limited by the number of photons that can be collected in each frame. Remote focusing has therefore mainly been used to achieve fast axial scanning in two-photon microscopy[9,10] and is not widely adopted as a method on the emission light path to refocus the detected image[11–13].

Here, we propose a remote focusing design that doubles the light efficiency, called FLIPR (flipped image remote focusing). FLIPR maintains the advantage of high-speed remote focusing but does not rely on polarization optics to separate the incoming from the refocused image. Instead of placing a mirror in the remote image plane, we use a microscopic retroreflector to flip and fold the image back into the remote focusing arm and spatially separate incoming and outgoing light (Fig. 1a–c). This design achieves volumetric imaging at up to 500 Hz for a z-range of 150 μm. We show its potential for enabling new high-speed volumetric imaging applications by recording the membrane potential of over a hundred neurons in the spinal cord of larval zebrafish in parallel.

## Results

To separate the refocused light from the incoming light in a remote focusing system, previous designs typically used a polarizing beamsplitter with a quarter-wave plate, which is light-inefficient due to the

[1]Einstein Center for Neurosciences, Charité – Universitätsmedizin Berlin, Berlin, Germany. [2]Present address: Université Paris Cité, Institute of Psychiatry and Neuroscience of Paris (IPNP), INSERM U1266 Paris, France. ✉e-mail: benjamin.judkewitz@charite.de

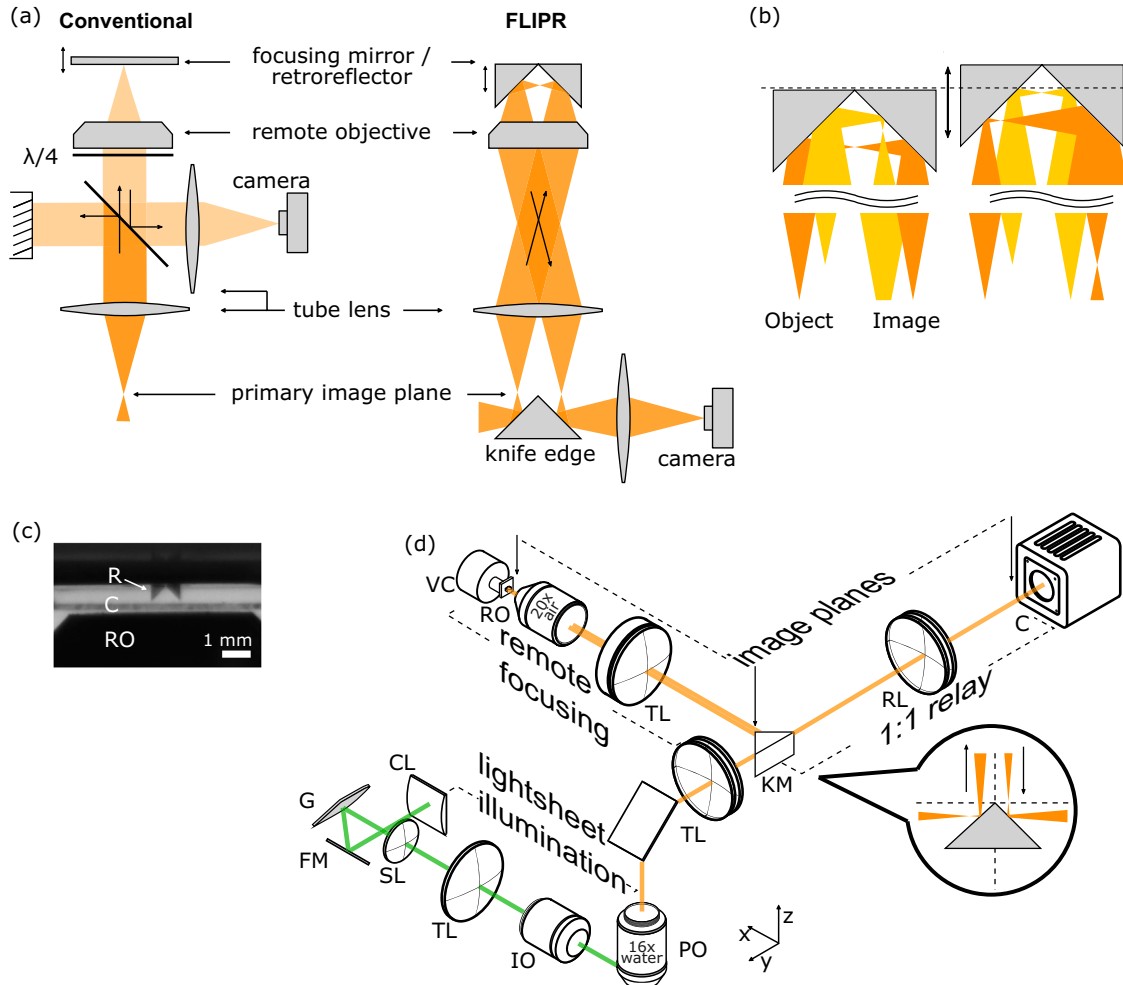

**Fig. 1 | Principle of the approach. a** Conceptual layout of our proposed design. Current polarizing beamsplitter based designs (left) lose half of the light intensity because unpolarized fluorescence is split into a beam dump and the remote focusing objective. Our design (right) uses half the available FOV for the incoming light and a retroreflector in the image plane of the remote objective to fold the image to the other side of the FOV. A knife edge mirror in the primary image plane is then able to direct the refocused light onto the camera. **b** Light path at the level of the object, retroreflector and image of two point sources located at different z-depths. Moving the retroreflector along the optical axis brings either of them in focus. **c** Picture of the microscopic retroreflector (R) together with the coverslip (C) and parts of the remote objective (RO). **d** Layout of the entire microscope for volumetric voltage imaging. A light sheet is generated with a cylindrical lens (CL) and projected onto the sample via a folding mirror (FM), galvo mirror (G), scan and tube lens (SL, TL) and illumination objective (IO). The galvo mirror controls the z-position of the light sheet. Fluorescence is collected via the primary imaging objective (PO) and imaged onto the knife edge mirror (KM) in the primary image plane and directed into the remote focusing path. The remote focusing path consists of a tube lens, remote objective (RO) and voice coil (VC) on which the retroreflector is mounted. After refocusing the image is relayed with a 1:1 relay (RL) onto the camera (C).

fact that fluorescence is unpolarized (Fig. 1a). FLIPR circumvents the need for polarization optics by making use of a miniature retroreflector. In addition to shifting the focal plane, the retroreflector flips half of the image from one side of the field of view (FOV) of the remote objective to the other. The refocused image can be picked off and relayed to the camera with a knife edge mirror in the conjugate image plane (Fig. 1b).

Based on this approach, we built a remote focusing system using a 16 × 0.8 numerical aperture (NA) water dipping primary objective and a 20 × 0.75 NA air remote objective. By using a non-standard focal length of 124 mm for the remote system tube lens, we achieved near-unit angular magnification of $M = 1.29$, close to the ideal $M = n_1/n_2 = 1.33$ (with $n_{1/2}$ being the respective refractive index of the immersion medium of the primary and remote objective)[8]. To achieve the proposed flipping of the image at the remote focusing plane, we custom-built a microscopic retroreflector from two 0.5 mm aluminum coated right-angle prisms. To determine the performance of our setup, we measured the point spread function (PSF) at multiple positions along

the z-axis and quantified lateral resolution as well as maximal intensity. Our setup achieves a maximal lateral resolution of 0.53 μm (theoretical limit: 0.34 μm) and maintains 80% of its maximal intensity over a z-range of -100 μm at the center of the FOV (Fig. 2a, b, Fig. S1).

Using a linear voice coil motor, we were able to move the retroreflector along the z-axis in the sample plane of the remote objective with up to 500 Hz over a range of 150 μm (Fig. 2c). This allowed us to access a volume of 388 μm in x-direction (limited by the size of the retroreflector) and 150 μm along the z-axis. The size of the y-axis is limited by the speed of the camera, essentially trading off FOV along the y-axis and number of planes in the volume (axes are referred to as shown in Fig. 1d). Due to inertial forces during high-speed motion of the voice coil motor, the retroreflector does not follow a strictly linear trajectory along the optical axis, but rather moves along an ellipsoid (Fig. S2ab). This results in a reproducible shift of ≤6.5 μm between the up- and the down-stroke along the y-axis of the image, which can be compensated digitally (perpendicular to the axis along which the image is flipped by the retroreflector, Fig. S2c). Due to the

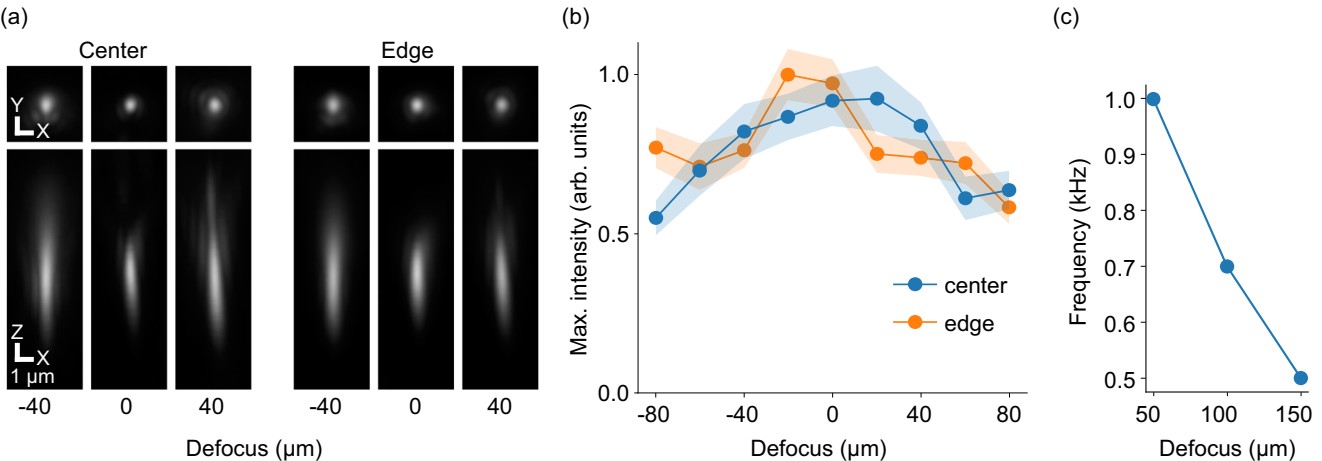

**Fig. 2 | Optical characterization. a** Averaged measured point spread functions from 0.1 μm diameter fluorescent beads over an 80 μm defocus range at the center and the edge of the FOV. **b** Normalized average maximal intensity from PSFs at each z-position. Intensity stays >80% over a range of ~100 μm in the center of the FOV and 60 μm at the edge. Shaded area designates standard error of the mean (s.e.m.). The PSF at each z-position in a and b was calculated by averaging over $n = 26–127$ beads. **c** Maximum drive frequency limited by the voice coil motor as a function of defocus.

continuously moving refocusing system and the rolling shutter of the camera, optical planes are also tilted in the volume relative to z-axis and in opposite directions during the down- and up-focus part of the volume (Fig. S3a and Fig. 3a). Both y-movement and tilt have to be taken into account for data analysis (Fig. S3b).

To showcase the performance of FLIPR for volumetric voltage imaging, we imaged the activity of spinal cord neurons in zebrafish larvae expressing the voltage indicator Voltron2 (*Tg(HuC:Gal4; UAS:-Voltron2-ST)*)[14]. We recorded 20 s of spontaneous activity in a volume spanning ~4 spinal segments (x = 390 μm, y = 46 um, z = 50 μm) divided into 2 × 8 planes (up-stroke and down-stroke) at 500 volumes/s. The data was then aligned and motion corrected and cell bodies were segmented in 3D to identify >200 cell bodies. After signal extraction and manual curation (see methods) the dataset contained meaningful signals from >100 neurons (Fig. 3b, c and Fig. S4). Since a single plane sweeps over a z-range of ~6.25 μm and with an average neuron size of $7.7 \pm 0.2$ μm, most neurons show signal in more than one imaging plane (see e.g. neuron 20 in plane 2 and 3 in Fig. 3b). Among the recorded neurons, some showed clear rhythmic membrane oscillations expected during fictive locomotor behavior, whereas others showed seemingly independent single spike activity (Fig. 3e, See also Video S1). Similar results were obtained in the hindbrain with an increased FOV and decreased number of frames per volume (Fig. S5). With the imaging conditions used in this study, we measured a bleaching of baseline fluorescence with a decay constant of $\tau = 72$ s (Fig. 3d), consequently we saw only minimal reduction of action potential SNR over the course of a 20 s recording (Fig. 3e).

## Discussion

Random access scanning[15] and multiplane confocal scanning[16] have recently attempted to expand voltage imaging to 3D volumes but remain limited by either the number of cells that can be recorded in parallel or the accessible volume owing to the limitations of point scanning approaches. Here we developed FLIPR, a technique that combines a light-efficient optical design for remote focusing with the advantages of high-speed camera-based readouts to potentially measure hundreds of neurons from large volumes. Since FLIPR does not rely on a polarizing beamsplitter, our design achieves a nominally twofold increase in light efficiency over comparable systems[13,17]. This is made possible by sacrificing half the FOV. In practice, this is not a limiting factor since with current camera technology the FOV is already reduced to achieve the kHz rates necessary for volumetric voltage

imaging. The principle of sacrificing FOV for efficiency was recently also proposed in[18], although geometrical constraints limit the attainable FOV to a small fraction of the full FOV height. Compared to a beamsplitter-based design, FLIPR does not increase the number of optical elements. However, since the knife edge mirror and the retroreflector are positioned in conjugate image planes, they are relatively sensitive to alignment errors. The retroreflector, while custom-built, can easily be assembled without special equipment (see methods for details).

Our spinal cord recordings were able to record spiking and oscillating neurons over the entire volume of the recorded spinal cord segments. This activity likely includes oscillating premotor interneurons as well as ventral and dorsal irregular spiking neurons[1]. Investigating the exact timing relation of these neurons during individual swimming episodes has previously been out of reach because imaging could only be done sequentially over multiple trials. It should be noted that the method described here is not limited to light sheet microscopy; rapid and light-efficient refocusing could also be used in widefield microscopy e.g. to provide volumetric data during voltage imaging from neurons of the mouse cortex or, at higher spatial resolution, of axons or dendrites distributed in a three-dimensional volume.

Depending on the nature of the sample, the volume from which we can record at a given rate is limited either by the camera speed or the maximal force output of the voice coil motor. For sparsely labeled samples, the refocusing can happen over a larger z-range within a single camera frame without projecting neurons on top of each other and thereby decreasing signal-to-noise. In such a situation the accessible volume is limited by the maximal range that the voice coil motor can scan over at a given frequency (which is limited by the maximal acceleration and thus the force output of the device). For relatively dense samples, where fine grained sampling along the z axis is necessary to avoid projecting cells on top of each other, the main limitation becomes the frame rate of the camera. In practice, this means that imaging other brain areas that require a larger FOV is possible either in sparsely labeled samples or, as is the case in the hindbrain recordings presented here, by reducing the size of the z-extent. This consideration also shows that the maximal number of recorded neurons will be approximately the same in both scenarios, as it is ultimately only a function of camera data rate.

Due to the 90° opening angle of the retroreflector and the unit angular magnification, our design limits the effective NA of the

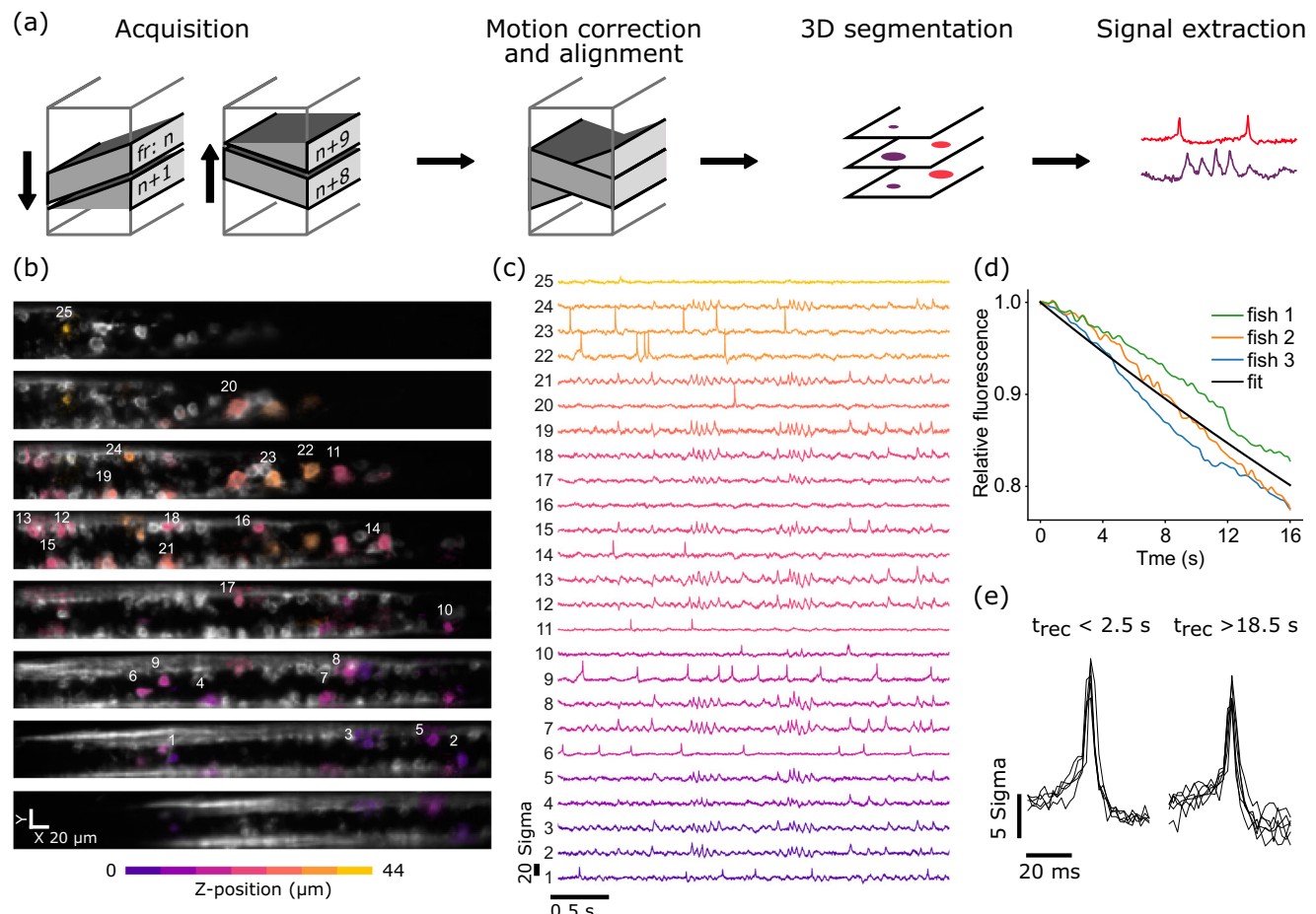

**Fig. 3 | High-speed volumetric voltage imaging. a** Schematic of the pipeline for volumetric voltage imaging data acquisition and signal extraction. 8 frames are acquired during the downward and another 8 during the upward stroke of the refocusing cycle. Frames at each z-position are then motion corrected in x-y and the time-average frame for each z-position is aligned with frames above and below to generate one single volume (see methods for details). This volume is then used for 3D segmentation. The obtained 3D ROIs are then used as the basis for the extraction of the fluorescent time traces (see methods for details). **b** Average fluorescence of 8 planes spanning the entire volume of 50 μm of spinal cord tissue of a 4 dpf zebrafish larva. animals are expressing UAS:Voltron2-ST under the control of HuC:Gal4 and are stained with JF-526 dye. Footprints of select neurons shown in (**c**).

Most footprints are visible over several z-sections but numbered only once. Shown is a representative example of an experiment with 4 fish. **c** Fluorescent traces corresponding to the spatial footprints shown in b (z-scored), ordered by z-position. Several neurons show clear single spikes while others show distinct oscillations likely corresponding to fictive swimming activity. Color of footprints in b and traces in c denote z-position in the volume. **d** Average bleaching of baseline fluorescence in recordings from 3 fish ($n_{cells}$ per fish = 28-114). **e** Closeup of several spikes of one example neuron (trace 6 in c) at the beginning (0 s –2.5 s) and end (18.5 s – 20 s) of the recording. Visible is the fast timing of individual spikes and only a small reduction in SNR at the end of the recording.

primary objective along one axis to NA < 0.7n (with *n* = refractive index of the immersion medium of the primary objective). This was not a limitation in the setup used here. Lastly, the FOV along the *x*-axis is currently limited by the size of the retroreflector. A custom non-square prism that is wider but not higher (to allow enough space to move within the limited working distance of the remote objective) would increase the FOV along the *x*-axis without compromising the FOV in y and z.

For technical simplicity, the illumination light in our setup stays constant over the entire period of focusing through the volume which comes with the downside of having to combine images from the up and down stroke phase of the volume scan. This could be avoided by doubling the laser intensity but shuttering the laser off during half the period, thus effectively only acquiring data from one time point during the volume scan. Analog modulation of laser intensity could also be used to correct for differences in z-speed (and thus integrated illumination intensity) of the light sheet at the center of the volume versus the turnaround points. This can also be mitigated by using a more powerful voice coil motor to better approach a triangle or sawtooth wave instead of the heavily filtered triangle/sinusoid intermediate used

in our current setup. With these improvements, we anticipate that FLIPR can be used to image hundreds of cells at kHz rate.

## Methods
### Microscope setup
Light Sheet illumination path: The light sheet was generated from a 532 nm laser (CNI MSL-FX-532), passed through a cylindrical lens (Thorlabs LJ1878L1-A, $f = 10$ mm) and beam reducer ($f1 = 100$ mm, $f2 = 75$ mm). Z-scanning was done with a galvo mirror (Cambridge Technology 6215H) before sending the beam through a scan lens ($f = 50$ mm), tube lens ($f = 400$ mm), and objective (Olympus XFluor 4X, NA 0.28). This resulted in a nominal Rayleigh length of the light sheet of 17 μm and $1/e^2$ width of 3.4 μm. To eliminate low-amplitude, high-frequency intensity fluctuations from the galvo position, focal length of all lenses was chosen to maximize the angle the galvo has to move to generate a given z-movement of the light sheet while still maintaining the desired width and Rayleigh length. The imaging chamber was custom-build from acrylic and mounted under the imaging objective. A cover slip was glued to one side to create a window for the excitation light to enter.

Detection and remote focusing path: Fluorescence was detected with a 16×0.8 NA water immersion objective (Nikon LWD 16x, NA 0.8) and filtered with a longpass filter to reject light from the excitation laser (Chroma ET542LP). The primary tube lens consisted of a 200 mm Plössl lens made from two 400 mm achromats (Thorlabs AC508-400-A). At the primary imaging plane the light was sent into the remote focusing arm using a knife edge mirror. The remote focusing arm consisted of a secondary tube lens made of a 180 mm tube lens (Thorlabs TTL-180) and an additional 400 mm achromat (Thorlabs AC508-400-A) to achieve an effective focal length of 124 mm. As a remote secondary objective we used a 20 × 0.7 NA air objective with a coverslip (Nikon Plan Apo 20x, 0.7 NA, WD = 1 mm) to achieve an overall magnification of M = 1.29, close to the ideal $M = n_1/n_2 = 1.33$. The microscopic retroreflector was custom-built from two aluminum coated 45 degree prism mirrors (Tower Optical 4531-0020). The prisms were carefully positioned opposite each other on a 5 mm x 5 mm piece of acrylic. A small amount of UV-curable glue (Panacol Vitralit 7041) was then delivered to the side of the prisms with a µl-pipette and hardened. The entire assembly was then mounted on a linear voice coil motor (Rapp Optoelectronic GLP-V1). After refocusing, the image was relayed with two 300 mm lenses (Thorlabs AC508-300-A) in Plössl configuration onto a high-speed sCMOS camera (Teledyne Photometrics Kinetix).

To assess the maximal optical performance, the setup was simulated in Zemax OpticStudio 17.5 to determine the diffraction limited FOV and with code published in ref. [19] to simulate the remote focusing range.

## Data acquisition

The camera was controlled using the manufacturer's proprietary software (PVCamTest 3.12.337) while all other waveforms to control the voice coil motor and galvo mirror were generated with custom python code.

The alignment and synchronization between light sheet and remote focusing system was calibrated by roughly defining the top and bottom position of both light sheet and focusing system in the volume of interest to get a rough initial alignment. The remote focus was then positioned at 10 equidistant focal planes in that volume and the 31 pictures with varying light sheet position around that focal plane were acquired. The ideal light sheet position was then determined by finding the plane with the highest contrast as described here[20] and linear regression was used to find the final relationship between light sheet and galvo position. During high-speed recordings we noticed a slight phase lag of the voice coil motor relative to the galvo mirror that led to some planes being out of focus. This was corrected by adding a manually determined phase shift. The galvo and voice coil motor were driven with a 500 Hz triangle wave to ensure the maximum time of linear motion through the volume. To avoid too much strain on the voice coil motor, the waveform was low pass filtered at 2000 Hz. The camera was run in 'frame-overlap trigger' mode where the exposure time of each frame is determined by the high state of an incoming trigger pulse to ensure that each frame would expose during the same phase of the volume scan.

Data was acquired at 500 Hz volume rate with 16 frames per period (8 frames per sweep through the volume). The camera was therefore triggered at 8 kHz.

## Optical characterization

To acquire point spread functions (PSF), the main camera was replaced by a camera (Basler daA3840-45 um) with smaller pixel size to better sample the PSF at the diffraction limit. Stacks of 0.1 µm fluorescent beads suspended in 1.5% agarose were acquired (pylon Viewer 5.1.0.12681) and a Gaussian was fitted to the x-, y-, and z-projection of each bead. The mean position of the fitted Gaussian was used to center an upsampled version of the bead image before averaging individual PSFs within a $64 \times 67 \times 12.6\ \mu m^3$ volume at either the center or the periphery of a 346 µm x 67 µm FOV. The peak fluorescence reported in Fig. 2b is the average peak value of the Gaussian fit to the x-profile of each bead at each z-position.

## Larval zebrafish voltage imaging

Animal husbandry followed national animal welfare guidelines and was approved by the Berlin LaGeSo authority. Three days post fertilization (dpf) *Tg(HuC:Gal4; UAS-Voltron2-ST)* zebrafish larvae were stained for at least 2 h in a solution of 3 µM Janelia Fluor 532 dye + 3% DMSO in fish facility water, rinsed twice and left to wash out remaining dye for at least 1 h. Three to four dpf larvae were paralyzed by immersion in 1 mg/ml α-bungarotoxin for 2–3 min and mounted in 1.5% low melting point agarose. To increase overall activity in the nervous system, the imaging chamber contained 20 mM pentylenetetrazole (PTZ). Maximal laser intensity after the light sheet objective was 15 mW.

## Data processing

Due to a mismatch between the DAQ clock generating the frame trigger and the camera internal clock, we noticed a jitter in the exposure time of each frame that led to small changes in full frame pixel counts as well as changes to baseline pixel values of each line which introduced additional noise. We first detected the timing of these changes and corrected for the relative change in fluorescence due to exposure time differences as well as additive changes in baseline for each pixel row (Fig. S6).

Time series for single planes were then motion corrected using the motion correction module included in the CaImAn package[21].

To generate a single 3D volume to draw regions of interest, the average fluorescence of each plane was first aligned to its adjacent planes. The rolling shutter nature of the camera results in a slightly tilted plane relative to the optical axis and the tilt is in the opposite direction when moving through the volume from top to bottom as compared to from bottom to top. To combine all planes in one single 3D stack for segmentation, each plane was therefore divided in half and stitched with the other half of a plane most closely matching its actual z-position (see Fig. S3b for an illustration).

The resulting 3D stack was then automatically segmented to generate 3D regions of interest (ROI) using cellpose and a custom-trained model[22].

To extract fluorescent time traces from the volume, we adapted the method described by Cai et al.[23]. to be used with 3D data instead of single planes. Briefly, this method estimates an initial spike train from the average fluorescence of the ROI and a background signal from a region around the ROI. Multiple iterations then produce a weighted sum of pixel values to only include fluorescence that contributes to the signal from a given cell. All traces and spatial footprints were manually checked to remove obvious duplicates and traces without neural activity.

## Reporting summary

Further information on research design is available in the Nature Portfolio Reporting Summary linked to this article.

## Data availability

The raw data used in this study are available at https://gin.g-node.org/danionella/Bohm_et_al_2024.

## Code availability

All custom code is available at https://github.com/danionella/Bohm_et_al_2024.

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

## Acknowledgements

We thank C. Berlage for initial discussion of the optical layout, M. Hoffmann, V. Cook, M. Kadobianskyi, J. Veith and C. Berlage for critically reading the manuscript. We thank E. Schreiter for sharing the UAS:Voltron2-ST line and M. Renz, A. Wrana and N. Kroworz for fish care. Data analysis was performed on the Berlin Institute of Health high-performance compute cluster. We acknowledge support by the German Research Foundation (DFG, projects EXC-2049-390688087 and 432195732) the European Research Council (ERC2016-StG-714560, ERC2021-CoG-101043615), the Einstein Foundation (EPP-2017-413), and the Alfried Krupp von Bohlen und Halbach Foundation.

## Author contributions

U.L.B. and B.J. conceived the project. U.L.B. designed and built the setup, performed experiments, and analyzed data. B.J. supervised the project. U.L.B. and B.J. wrote the manuscript.

## Funding

## Competing interests

The authors declare no conflicts of interest.
