## [Transparent Peer Review file · Nature Communications]

Fast and light efficient remote focusing for volumetric voltage imaging

Corresponding Author: Professor Benjamin Judkewitz

Version 0:

Reviewer comments:

Reviewer #1

(Remarks to the Author)

Bohm et al. introduced an improved version of the remote focusing technique by using FLIPR approach to light sheet microscopy. Despite the weakness of losing FOV, they achieved volumetric voltage imaging of over 100 spinal cord neurons at 500 volumes/s using voltron2 GEVI in zebrafish larvae. Optical recording of membrane potentials is an important technique for spatiotemporal analysis of neuronal populations and other cell types; thus, this technical advance is valuable. However, the quality of the data presented and the validation of their system are not sufficient. Thus, improvements in the manuscript are required.

- In FLIPR voltage imaging, images acquired by upward and downward strokes need to be combined to obtain each z-slice image. Thus, each pixel in the obtained images contains information on multiple time points. This may be problematic for the detailed analysis of spatiotemporal dynamics of neurons. Please describe the extent to which temporal information is lost during this imaging processing (depending on imaging conditions), as well as possible influences and solutions.

- The authors wrote in the abstract and result sections that "remote focusing doubles optical efficiency," but such data was not provided. Please provide evidence, for instance, by comparing the results of the previous remote focusing systems with those of the newly developed one.

- Regarding the PSF of this new system, how about the uniformity of PSF at different XY locations? This information is important to evaluate the correctness of the obtained images. For instance, please discuss PSF at the center and peripheral regions of FOV. Also, it is preferable to show the fitted curves next to the PSF images.

- In this manuscript, the authors did not present enough data to evaluate in vivo voltage imaging results. For instance, compare SNR and kinetics of voltage dynamics of spinal cord neurons with the ones obtained by conventional light sheet microscopy and show the superiority of the current system.

What percentage of the spinal cord neurons labeled in Tg(HuC:Gal4; UAS-Voltron2-ST) line were recorded in this voltage imaging? Is this sufficient for analyzing circuit dynamics?

The authors succeeded in volumetric imaging of eight z-steps for a depth of 50 μm . Is this sufficient resolution according to the size of the spinal cord neurons? Please describe this with data in the manuscript.

- How stable is this recording? How many trials of recordings did you conduct using one fish? How many fish did you test in this study? Please provide information on the reproducibility of these results and possible phototoxicity.

- The authors should discuss the biological aspects of the spinal cord recordings obtained in this study by referring to previous studies.

- Did you obtain any new phenomena that were not detected by conventional light sheet microscope imaging? Or what do you expect based on the specifications of this new approach?

- Did you try this imaging approach in brain regions in zebrafish larvae? Please discuss in this manuscript to what extent

this technology is applicable, including possible limitations.

- Why did you use Voltron2 for this recording? If you have tested other sensors, it would be good to compare the results with them to show the significance of this sensor.
- Line103: The authors wrote "single spike activity," but they did not provide the evidence. Either electrophysiological recordings or at least kinematic analysis is required to conclude this.
- The overall quality of the Figures is not enough and needs improvement. The appropriate labels (eg. arrows and names) should be placed on the parts of the figures to be explained, including the optics, and if labels are placed, the explanations should be included in the Figure Legend.
- Please provide detailed information (+illustration) about the imaging chamber used for this voltage imaging.
- Figure S3b: Add more description for each Figure, such as z-position. This is also the case for Figure 3b. The scale bar and color information are also missing.

Reviewer #3

(Remarks to the Author)

Böhm and Judkewitz present a novel method of remote focusing that can be applied in the collection arm of a fluorescence light-sheet microscope. Their method uses a retroreflective element mounted to a voice coil actuator combined with a pickoff mirror to adjust the focal plane without polarization optics. This is advantageous as it avoids the light-loss normally associated with using polarization-based remote focusing schemes with (unpolarized) fluorescent light. The authors demonstrate their method in a light-sheet microscope for the application of high-speed voltage imaging, showing that they can achieve light-efficient imaging at 500Hz volumetric scan rates.

Voltage imaging is not my area of expertise, so I cannot speak to the impact for this specific application or to the strength of the voltage imaging methods / data presented here. From a microscopy standpoint, I find this to be an elegant solution which solves a significant and known problem in remote focusing. I agree that this method is applicable to a range of light-sheet and other microscopy systems and will be of interest to readers. The manuscript is well-written, and I support its publication in Nature Communications. A few minor suggestions are below:

The proposed method does not require polarization optics, which simplifies the optical design in some sense and increases light-efficiency as described. However, it does require assembly of a custom retroreflective element and precise alignment of the retroreflective element and the knife edge mirror. I imagine these elements could be difficult to manufacture and align. Could the authors comment on how difficult it is to build this setup compared to a traditional polarization-based system? I believe this would be interesting to readers who are considering implementing this design.

Line 79: it would be helpful to indicate the coordinate system (X, Y, Z) referenced here in a figure, such as Fig. 1d.

Fig. S2: the strategy to correct for rolling shutter artifacts by combining parts of the up and down sweep frames is clever, but it took me a while to understand from the figure. In panel a, it is hard to tell that the lower plots are square waves – perhaps make the plot extend beyond 1 and 0 so the trace edge does not intersect the plot edges? Or make the traces a different color? In panel b, I find it confusing that the first corrected frame (left side of 1 and right side of 4) is in the same position in the left and right diagrams, but the second corrected frame (left side of 4 and right side of 1) is shifted down words in the right diagram relative to the left diagram. In reality, the two corrected frames overlap – correct? Similarly, I find it confusing that the four frames at the left are converted to only two frames at the right – there should still be four frames post-correction, correct? I would rework this figure.

Line 92: the captions specifies "n ≥ 39-103 beads". I'm not sure what this means, can the authors clarify?

Line 108: the caption says "Average frames at each position are then motion corrected". It's unclear where averaging (arithmetic mean) is being used. Are the authors referring to the process shown in Fig. S2 of splitting frames and recombining them? If so, I would suggest a different term than "average". Otherwise please clarify.

Typos:

- Fig. S2 is mislabeled as S1
- Fig. S3 caption: trances > traces
- Line 198: missing model number
- Line 228: plains > planes

Version 1:

Reviewer comments:

Reviewer #3

(Remarks to the Author)

The authors have addressed all of my comments - this is a high-quality and interesting paper, and I support publication in Nature Communications. One more typo on line 201: Toweroptical > Tower Optical

Overall I find the responses adequate. Regarding the first few comments on the optical aspects of the paper:

- Temporal artifacts: I agree with the authors' response - conclusions should be limited to those slower than the volumetric scan rate, but this is the case for any scanning imaging system. I don't find this to be unique limitation. Combining multiple time points is a bit unique, but as the authors explain this behavior could be turned off if desired for certain applications by shuttering the laser.

- Remote focusing efficiency: I agree with the authors - the theoretical argument of doubled remote focusing efficiency due to collecting both polarization states is sufficient. I think experimental validation of light efficiency would be possible but complicated (quantifying light efficiency across different imaging setups in a controlled way is difficult in my experience). I don't find it necessary here as the theoretical explanation is straightforward and sound.

- PSF uniformity: I think this is a good point raised by the other reviewer. The authors respond by assessing uniformity based on the average PSF intensity. Quantifying average PSF dimensions (e.g. full width at half maximum) would be a little more robust in my opinion, but I think the response is still sufficient.

The other reviewer's remaining comments focused on the voltage imaging methods and results. I don't have any expertise in voltage imaging, so I can't speak to suitability of the remote focusing technique to this specific application or the robustness of the voltage imaging results presented. The voltage imaging should be done correctly of course. However in my view, the manuscript is primarily a demonstration of a novel optical technique, which I think is sound.

I agree with the authors' responses that any new biological discoveries are outside of the scope of this study. Similarly, I think characterization of different voltage sensors, brain regions, etc. is not really necessary for a proof of concept to demonstrate this optical technique, and I find the current treatment of these issues to be sufficient.

Ultimately I think voltage imaging is one example use case. This strategy could be useful in any situation requiring high-speed volumetric imaging (cell dynamics, embryo development, etc.) and I think any of these would have been a suitable example.

Reviewer #1 (Remarks to the Author):

Bohm et al. introduced an improved version of the remote focusing technique by using FLIPR approach to light sheet microscopy. Despite the weakness of losing FOV, they achieved volumetric voltage imaging of over 100 spinal cord neurons at 500 volumes/s using voltron2 GEVI in zebrafish larvae. Optical recording of membrane potentials is an important technique for spatiotemporal analysis of neuronal populations and other cell types; thus, this technical advance is valuable. However, the quality of the data presented and the validation of their system are not sufficient. Thus, improvements in the manuscript are required.

We thank the reviewer for their detailed and thorough comments. We hope to have responded to the remaining questions and sufficiently improved the presented data and figures as described below.

- In FLIPR voltage imaging, images acquired by upward and downward strokes need to be combined to obtain each z-slice image. Thus, each pixel in the obtained images contains information on multiple time points. This may be problematic for the detailed analysis of spatiotemporal dynamics of neurons. Please describe the extent to which temporal information is lost during this imaging processing (depending on imaging conditions), as well as possible influences and solutions.

We combined upward and downward stroke, but it is not necessary for the technique. Doubling the laser intensity but shuttering the laser off during half the period would lead to the overall same light exposure but only images from one time point. We don't anticipate any particular complications for data analysis by sampling twice from a given point at two timepoints within a single volume scan. No conclusions should be drawn for any timing that is faster than the nyquist criterion of the overall volume rate (4 ms in our case).

Potentially problematic could be the fact that light is collected from only a short moment during the 2 ms it takes to acquire each volume. It is therefore possible that events significantly shorter than 2 ms are not captured. Such aliasing is however a common problem in all acquisition techniques that involve scanning, be it virtual lightsheets with a scanned pencil beam or point scanning techniques. To make this more clear in the manuscript, we rewrote the sentence in line 171-174 to clarify how the data acquisition and therefore analysis could be simplified by shuttering the laser.

It now reads: *“For technical simplicity, the illumination light in our setup stays constant over the entire period of focusing through the volume which comes with the downside of having to combine images from the up and down stroke phase of the volume scan. This could be avoided by doubling the laser intensity but shuttering the laser off during half the period, thus effectively only acquiring data from one time point during the volume scan.”*

The authors wrote in the abstract and result sections that "remote focusing doubles optical efficiency," but such data was not provided. Please provide evidence, for instance, by comparing the results of the previous remote focusing systems with those of the newly developed one.

We now specify in the abstract that the claim about doubling light efficiency is *“compared to conventional beamsplitter-based techniques”*(line 10) which, in its previously published design, can only utilize one of the two polarizations and thus, at most, half of the emitted light. We also now

emphasize this more clearly in the discussion (lines 135-137) and added citations to two other studies that used this less efficient design for comparison. It now reads: “*Since FLIPR does not rely on a polarizing beamsplitter, our design achieves a nominally two-fold increase in light efficiency over comparable systems*”

• Regarding the PSF of this new system, how about the uniformity of PSF at different XY locations? This information is important to evaluate the correctness of the obtained images. For instance, please discuss PSF at the center and peripheral regions of FOV. Also, it is preferable to show the fitted curves next to the PSF images.

To better evaluate the PSF at the center vs the periphery, Fig 2a now includes average PSFs of a 64 μm x 67 μm square in the center and an equal square at the periphery as well as the maximum intensity at each z-position for the center vs the edge in Fig. 2b. We also added a new supplementary figure 1 which shows an extended selection of PSFs over the same z-range as well as their respective cross sections to better evaluate the quality of the PSF over the entire imaging volume.

In this manuscript, the authors did not present enough data to evaluate in vivo voltage imaging results. For instance, compare SNR and kinetics of voltage dynamics of spinal cord neurons with the ones obtained by conventional light sheet microscopy and show the superiority of the current system.

In the presented work we do not claim superior SNR. We rather claim that this is the first time that voltage was measured in 3D with light sheet microscopy. Voltage imaging with conventional light sheet microscopy has so far only been measured from a single optical plane at the time. The possibility to measure from a volume instead of a single plane and not the quality of the data as such constitutes the superiority of the system. We went through the manuscript and confirmed that all our claims are consistent with this statement.

What percentage of the spinal cord neurons labeled in *Tg(HuC:Gal4; UAS-Voltron2-ST)* line were recorded in this voltage imaging? Is this sufficient for analyzing circuit dynamics?

The *Tg(Huc:Gal4; UAS-Voltron2)* line used in this study labels a random subset of all the neurons in the zebrafish brain. Given that our FOV covers roughly 3-4 spinal cord segments and there are 30 segments in total, our volume covers about 10% of the spinal cord neurons. These segments are repeating elements and imaging a few likely is a good approximation of what happens elsewhere. The main variation between cell types happens along the z-axis as can be seen by the different firing dynamics at different z-positions in the volume (fig. 3c). In line 144-151 we now discuss more explicitly the possible insights that can be gained from such recordings. It should also be noted that due to the constraints in camera speed the total FOV (x, y and z) is not smaller than other approaches but we extend the FOV along the z direction to record a volume by reducing its size along y.

The authors succeeded in volumetric imaging of eight z-steps for a depth of 50 μm . Is this sufficient resolution according to the size of the spinal cord neurons? Please describe this with data in the manuscript.

We analyzed the size of the recorded neurons to emphasize that “... a single plane sweeps over a z-range of approximately $6.25\ \mu\text{m}$ and with an average neuron size of $7.7 \pm 0.2\ \mu\text{m}$, most neurons show signal in more than one imaging plane” (lines 104-106). It should also be noted that since the camera is constantly exposing while the light sheet moves through the volume, neurons are not lost when recording fewer slices but are instead projected on top of each other within a single frame.

How stable is this recording? How many trials of recordings did you conduct using one fish? How many fish did you test in this study? Please provide information on the reproducibility of these results and possible phototoxicity.

The reviewer raises an important point. We included panel Fig. 3d and state that “*With the imaging conditions used in this study, we measured a bleaching of baseline fluorescence with a decay constant of $\tau = 72\ \text{s}$ (Fig. 3d), consequently we saw only minimal reduction of action potential SNR over the course of a 20 s recording (Fig 3e).*” (lines 110-112) to report the baseline fluorescence bleaching in 3 fish and a visual comparison of spike SNR at the beginning and end of a recording. The measured fluorescence decay constant of 72 s suggests recordings are possible for roughly 1 minute which is on the same order of magnitude as previously reported times for voltage imaging in spinal cord neurons (Böhm *et al.* 2021)

The authors should discuss the biological aspects of the spinal cord recordings obtained in this study by referring to previous studies.

Did you obtain any new phenomena that were not detected by conventional light sheet microscope imaging? Or what do you expect based on the specifications of this new approach?

The spinal cord recordings in this study were mainly done as a proof of principle to showcase the technology and any biological discoveries were beyond the scope of this study. We agree however that it is important to the reader to understand the potential benefits and applications for zebrafish voltage imaging and beyond. We therefore supplemented our discussion to explain that “*Our spinal cord recordings were able to record spiking and oscillating neurons over the entire volume of the recorded spinal cord segments. This activity likely includes oscillating premotor interneurons as well as ventral and dorsal irregular spiking neurons. Investigating the exact timing relation of these neurons during individual swimming episodes has previously been out of reach because imaging could only be done sequentially over multiple trials.*” (lines 144-148). We also mention potential application beyond zebrafish voltage imaging by stating that “*It should be noted that the method described here is not limited to light sheet microscopy; rapid and light efficient refocusing could also be used in widefield microscopy e.g. to provide volumetric data during voltage imaging from neurons of the mouse cortex or, at higher spatial resolution, of axons or dendrites distributed in a three-dimensional volume.*” (lines 148-151)

Did you try this imaging approach in brain regions in zebrafish larvae? Please discuss in this manuscript to what extent this technology is applicable, including possible limitations.

To give a concrete example of how this technology can be used in other brain regions of the larval zebrafish, we now include supplementary figure S4 showing recordings from the zebrafish hindbrain. This also serves as an example of a recording with an increased FOV along the y-axis and reduced z-extend. We discuss the tradeoffs between FOV and z-depth by stating that “*In practice this means that imaging other brain areas that require a larger FOV is possible either in sparsely labeled*

samples or, as is the case in the hindbrain recordings presented here, by reducing the size of the z-extent.” (lines 159-161).

Why did you use Voltron2 for this recording? If you have tested other sensors, it would be good to compare the results with them to show the significance of this sensor.

Voltron2 together with the JF-526 dye was used because it is one of the most commonly used dyes for 1-photon excitation at standard laser wavelengths as well as the availability of transgenic animals. We did not compare Voltron2 with other sensors, as this is not the subject of our paper and has previously been done (e.g. Abdelfattah *et al.* 2023 for comparison of Voltron1 and Voltron2 in zebrafish).

Line103: The authors wrote "single spike activity," but they did not provide the evidence. Either electrophysiological recordings or at least kinematic analysis is required to conclude this.

We added Fig. 3e to show single spike activity on the millisecond timescale similar to what has been shown in previous studies using the same sensor (Abdelfattah *et al.* 2023)

The overall quality of the Figures is not enough and needs improvement. The appropriate labels (eg. arrows and names) should be placed on the parts of the figures to be explained, including the optics, and if labels are placed, the explanations should be included in the Figure Legend.

- Please provide detailed information (+illustration) about the imaging chamber used for this voltage imaging.
- Figure S3b: Add more description for each Figure, such as z-position. This is also the case for Figure 3b. The scale bar and color information are also missing.

We thank the reviewer for the suggestions and added additional information to the legend of Fig. S3, better labeling of the optics in Fig. 1 and a reference to the details of the custom build imaging chamber.

Reviewer #3 (Remarks to the Author):

Böhm and Judkewitz present a novel method of remote focusing that can be applied in the collection arm of a fluorescence light-sheet microscope. Their method uses a retroreflective element mounted to a voice coil actuator combined with a pickoff mirror to adjust the focal plane without polarization optics. This is advantageous as it avoids the light-loss normally associated with using polarization-based remote focusing schemes with (unpolarized) fluorescent light. The authors demonstrate their method in a light-sheet microscope for the application of high-speed voltage imaging, showing that they can achieve light-efficient imaging at 500Hz volumetric scan rates.

Voltage imaging is not my area of expertise, so I cannot speak to the impact for this specific application or to the strength of the voltage imaging methods / data presented here. From a microscopy standpoint, I find this to be an elegant solution which solves a significant and known problem in remote focusing. I agree that this method is applicable to a range of light-sheet and other microscopy systems and will be of interest to readers. The manuscript is well-written, and I support its publication in Nature Communications. A few minor suggestions are below:

We thank the reviewer for their positive comments and helpful suggestions. We implemented them in the revised manuscript as described below.

The proposed method does not require polarization optics, which simplifies the optical design in some sense and increases light-efficiency as described. However, it does require assembly of a custom retroreflective element and precise alignment of the retroreflective element and the knife edge mirror. I imagine these elements could be difficult to manufacture and align. Could the authors comment on how difficult it is to build this setup compared to a traditional polarization-based system? I believe this would be interesting to readers who are considering implementing this design.

We did not directly compare building our system and a polarization based system back to back and the number of optical elements are actually the same, the reviewer correctly points out that the knife edge mirror and retroreflector are in critical positions and therefore sensitive to alignment errors. We now mention in the discussion that “*Compared to a beamsplitter-based design, FLIPR does not increase the number of optical elements. However, since the knife edge mirror and the retroreflector are positioned in conjugate image planes, they are relatively sensitive to alignment errors . The retroreflector, while custom built, can easily be assembled without special equipment (see methods for details).*” (lines 139-143). To facilitate implementing the custom built retroreflector in other labs, we describe the construction in more detail in the methods (lines 200-202). Building the retroreflector does not need any special equipment and we anticipate that any lab that is capable of building a beam splitter based system should not have any major difficulty implementing our design.

Line 79: it would be helpful to indicate the coordinate system (X, Y, Z) referenced here in a figure, such as Fig. 1d.

As suggested, we added a coordinate system in Fig. 1d to better understand which axes are referred to in the text.

Fig. S2: the strategy to correct for rolling shutter artifacts by combining parts of the up and down sweep frames is clever, but it took me a while to understand from the figure. In panel a, it is hard to tell that the lower plots are square waves – perhaps make the plot extend beyond 1 and 0 so the trace edge does not intersect the plot edges? Or make the traces a different color? In panel b, I find it confusing that the first corrected frame (left side of 1 and right side of 4) is in the same position in the left and right diagrams, but the second corrected frame (left side of 4 and right side of 1) is shifted down words in the right diagram relative to the left diagram. In reality, the two corrected frames overlap – correct? Similarly, I find it confusing that the four frames at the left are converted to only two frames at the right – there should still be four frames post-correction, correct? I would rework this figure.

The reviewer is correct that the corrected frames overlap. We agree that the schematics in this figure were unclear. We reworked the entire figure S2 and hope it is visually more intuitive.

Line 92: the caption specifies “ $n \geq 39-103$ beads”. I’m not sure what this means, can the authors clarify?

The PSF at each position was calculated from the average of a certain number of fluorescent bead measurements at each given z-position. In the figure legend we now write that “*The PSF at each z-position was calculated by averaging over $n = 26-127$ beads.*” (lines 95-96) to make this clearer.

Line 108: the caption says “Average frames at each position are then motion corrected”. It’s unclear where averaging (arithmetic mean) is being used. Are the authors referring to the process shown in Fig. S2 of splitting frames and recombining them? If so, I would suggest a different term than “average”. Otherwise please clarify.

The phrasing was indeed imprecise. Recordings are first motion corrected in x-y and then the arithmetic mean for each plane over the entire recording are aligned in z as shown in Fig. S2. We now write that “*Frames at each z-position are then motion corrected in x-y and the time-average frame for each z-position is aligned with frames above and below to generate one single volume (see methods for details). This volume is then used for 3D segmentation.*” (lines 117-119)

Typos:

Fig. S2 is mislabeled as S1

Fig. S3 caption: trances > traces

Line 198: missing model number

Line 228: plains > planes

We thank the reviewer for spotting these typos, they have been corrected.